# In vivo CRISPR/Cas9 targeting of fusion oncogenes for selective elimination of cancer cells

M. Martinez-Lage [1,10], R. Torres-Ruiz [1,2,10✉], P. Puig-Serra [1], P. Moreno-Gaona[1], M. C. Martin[1], F. J. Moya[1], O. Quintana-Bustamante [3,4], S. Garcia-Silva [5], A. M. Carcaboso [6,7], P. Petazzi[2], C. Bueno[2], J. Mora [6,7], H. Peinado [5], J. C. Segovia [3,4], P. Menendez[2,8,9] & S. Rodriguez-Perales [1✉]

Fusion oncogenes (FOs) are common in many cancer types and are powerful drivers of tumor development. Because their expression is exclusive to cancer cells and their elimination induces cell apoptosis in FO-driven cancers, FOs are attractive therapeutic targets. However, specifically targeting the resulting chimeric products is challenging. Based on CRISPR/Cas9 technology, here we devise a simple, efficient and non-patient-specific gene-editing strategy through targeting of two introns of the genes involved in the rearrangement, allowing for robust disruption of the FO specifically in cancer cells. As a proof-of-concept of its potential, we demonstrate the efficacy of intron-based targeting of transcription factors or tyrosine kinase FOs in reducing tumor burden/mortality in in vivo models. The FO targeting approach presented here might open new horizons for the selective elimination of cancer cells.

[1] Molecular Cytogenetics and Genome Editing Unit, Human Cancer Genetics Program, Centro Nacional de Investigaciones Oncológicas (CNIO), 28029 Madrid, Spain. [2] Josep Carreras Leukemia Research Institute and Department of Biomedicine, School of Medicine, University of Barcelona, 08036 Barcelona, Spain. [3] Differentiation and Cytometry Unit, Hematopoietic Innovative Therapies Division, Centro de Investigaciones Energéticas, Medioambientales y Tecnológicas (CIEMAT), Centro de Investigaciones Biomédicas en Red de Enfermedades Raras (CIBERER), 28040 Madrid, Spain. [4] Advanced Therapies Mixed Unit, Instituto de Investigación Sanitaria-Fundación Jiménez Díaz (IIS-FJD, UAM), 28040 Madrid, Spain. [5] Microenvironment and Metastasis Group, Molecular Oncology Program, Spanish National Cancer Research Centre, 28029 Madrid, Spain. [6] Institut de Recerca Sant Joan de Deu, Barcelona, Spain. [7] Department of Pediatric Hematology and Oncology, Hospital Sant Joan de Deu, 08950 Barcelona, Spain. [8] Instituciò Catalana de Recerca i Estudis Avançats (ICREA), Passeig Lluis Companys, 08010 Barcelona, Spain. [9] Centro de Investigación Biomédica en Red de Cáncer (CIBER-ONC), ISCIII, Barcelona, Spain. [10] These authors contributed equally: M. Martinez-Lage and R. Torres-Ruiz. ✉email: rtorresr@cnio.es; srodriguezp@cnio.es

Despite advances in treatment, cancer continues to cause over nine million deaths worldwide each year. Indeed, there is still much to be learned about the biologic behavior of cancer cells and molecular disease mechanisms. In this sense, genome editing offers unprecedented opportunities to further our knowledge of cancer biology and to fuel the development of new preclinical models and more efficient and directed strategies to eliminate cancer cells[1]. Unlike other genetic diseases such as Duchenne muscular dystrophy or cystic fibrosis, cancer development involves several genetic mutations that can deregulate multiple genes. In the context of cancer gene therapy, it is clear that targeting a single gene is often insufficient to eliminate cancer cells—yet, many types of cancers are addicted to the presence of a single oncogenic event that can reprogram cells by deregulating downstream molecular and (epi)-genetic programs[2] and initiate tumorigenesis. This is the case for the so-called fusion oncogenes (FOs), which are chimeric genes resulting from in-frame fusions of the coding sequences of two genes involved in a chromosomal rearrangement[3]. While the nature of the FOs may be diverse, they are primarily classified as involving transcription factors or tyrosine kinases. Silencing of FO transcripts has been shown to inhibit tumor cell growth in vitro and in vivo[4,5], demonstrating FO addiction in many human cancers. Moreover, genome-wide mutational studies—particularly in childhood cancers—have provided additional support for some FOs as the sole necessary genetic drivers for cancer initiation by revealing silent mutational landscapes in FO-initiated tumors[6,7].

FOs are recurrent genomic findings in human cancer and are characterized by patient-specific genomic breakpoints that occur in intronic regions, rarely disrupting coding sequences. Analysis of data from The Cancer Genome Atlas suggests that FOs drive the development of more than 16% of human cancers[8], including mesodermal cancers (typically leukemias, lymphomas and sarcomas). FOs have also been found in epithelial cancers, including prostate[9], colorectal[10], breast[11] or melanoma[12], and over 350 recurrent FOs involving >300 different genes have been identified to date[13,14]. Given their restriction to cancer cells, FOs are attractive targets for directed therapy; however, therapeutic targeting of specific FOs has remained challenging due to the difficulties in specifically recognizing and targeting the resultant chimeric protein, and also because FO products are intracellular, necessitating effective intracellular approaches for delivery of therapeutic molecules targeting the chimeric transcripts/proteins[15]. Along this line, small molecules[16], intrabodies[17] and aptamers[18] have been used successfully to target fusion proteins, and antisense RNA[19], ribozymes[20] and RNAi[21] to target the fusion transcripts. Likewise, the development of genome editing approaches offers new possibilities to directly target and modify the genomic sequence of cancer cells[22].

Clustered Regularly Interspaced Short Palindromic Repeats (CRISPR)/Cas9-based technology has revolutionized genome editing of mammalian cells, and can generate targeted breaks at any desired location in the genome—opening new horizons for therapeutic gene editing to correct monogenic or somatic mutations[23,24]. CRISPR/Cas9 induced breaks can be repaired by one of two major double-strand break (DSB) repair pathways: the inefficient but error-free homology-directed repair (HDR) pathway, which requires a DNA template; and the highly efficient but error-prone non-homologous end joining (NHEJ) pathway, which does not require a DNA template. A recent pioneering study by Chen et al. exploited the HDR repair pathway to insert a suicide gene at patient-specific breakpoints of FOs in prostate and hepatocellular cancer cell lines[25]. Although very elegant, the combination of an extremely low efficiency (1–10%) HDR-based approach[26] and the requirement of patient-specific breakpoint

CRISPR tool development limits the utility of this strategy for developing targeted cancer therapies.

In the present study, we report an efficient NHEJ CRISPR-mediated genome editing strategy for targeting FOs, which we believe represents a valid approach for directed elimination of cancer cells harboring a given FO. The approach is based on targeting two intronic sequences—one in each of the genes involved in the FO—that induces a cancer cell-specific genome deletion that eliminates key protein domains or changes the reading frame of the FO. Notably, this gene editing-based approach induces deletion only in cells harboring a FO without affecting exonic sequences or protein expression of the germline non-rearranged alleles. The same two single guide RNAs (sgRNAs) allow the targeting of different isoforms or every patient-specific breakpoint of a given FO, and is thus a universal approach for cancer-associated FOs. In vitro analysis and patient-derived xenograft (PDX) models show that delivery of CRISPR/Cas9 components targeting a FO results in efficient and specific tumor growth control. Our findings provide a proof-of-concept for a highly efficient and non-breakpoint-specific genome editing strategy-targeting FOs as an innovative approach for selective elimination of cancer cells.

## Results

**CRISPR-mediated intronic targeting enables FO disruption.** Our rationale was to devise a genome editing approach to specifically disrupt FOs in cancer cells that fulfils two strict criteria: i) it would not affect the exonic sequences or the expression of wild-type alleles involved in the rearrangement, and ii) it would be feasible irrespective of the FO isoform or the patient-specific breakpoint. To test this approach, we first used a cellular model of Ewing sarcoma, one of the most common cancers in children/adolescents characterized by a chromosomal translocation that fuses the transactivation domain of the RNA-binding protein EWSR1 to the DNA-binding domain of an ETS protein, most commonly FLI1. The EWSR1-FLI1 (EF) fusion acts as a dominant transcription factor, and cells are addicted to EF expression[27–29]. Two main *EF* isoforms exist that fuse exon 7 of *EWSR1* to either exon 5 (*EF* type 2) or exon 6 of *FLI1* (*EF* type 1)[30–32].

We designed a strategy to induce *EF*-specific genomic deletion targeting two genomic introns—one in each rearranged gene – flanking the breakpoint introns. To target all isoforms of *EF*, four sgRNAs were designed to target introns 3 and 6 of *EWSR1* and introns 6 and 8 of *FLI1* (Fig. 1a and Supplementary Table 1). sgRNAs were designed specifically to not disrupt described splice acceptor or donor sites or transcription regulators such as enhancers or silencers. We also confirmed that the sgRNA target sites were not affected by common single nucleotide polymorphisms (SNPs) (Supplementary Fig. 1a, b). Targeted introns were selected to generate large deletions including key functional domains of the FO, to induce a frameshift event in the remaining 3′ region of the *FLI1* gene, and to cover all the common hotspot introns within the break cluster regions. Consequently, genomic deletions will occur only in cells harboring the FO with both on-target intronic regions in the same chromosome. Crucially, intron-directed sgRNAs guarantee the germline configuration of non-rearranged *EWSR1* and *FLI1* alleles, such that the expression of wild-type alleles is preserved in healthy cells.

Using a single sgRNA lentiviral expression vector (pLV-U6sgRNA-EFSCas9)[33], we tested the efficiency of genomic deletion with four combinations of sgRNAs (sgE3-sgF6, sgE3-sgF8, sgE6-sgF6, sgE6-sgF8) in the A673 Ewing sarcoma cell line. Sanger sequencing analysis of PCR products using oligonucleotides flanking the targeted loci confirmed genomic deletions (Supplementary Fig. 2a), and EF-targeted A673 cells showed a

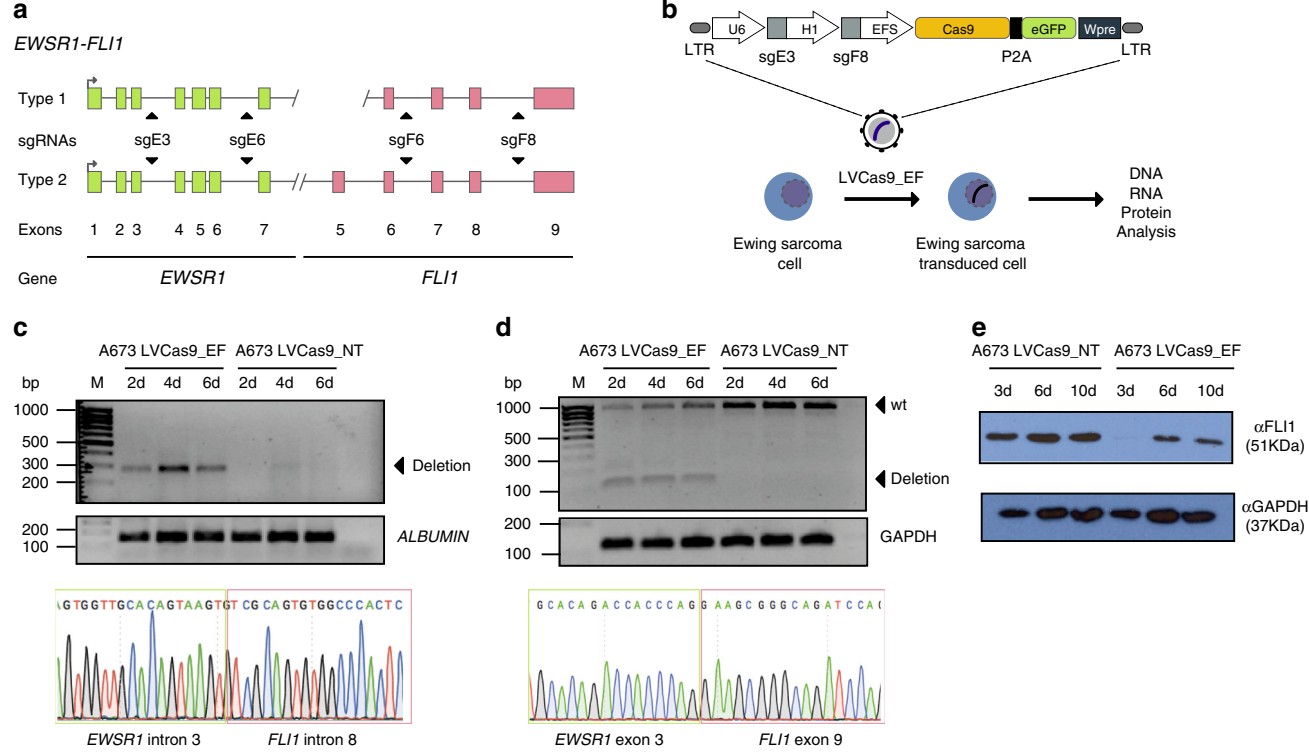

**Fig. 1 Strategy and in vitro CRISPR-mediated disruption of EWSR1-FLI1. a** Scheme representing type 1 and 2 *EWSR1-FLI1* FOs, illustrating the genomic structure with exon arrangement and sites of fusion. sgRNAs targeting introns 3 and 6 of *EWSR1* and 6 and 8 of *FLI1* are indicated. **b** Schematic representation of the all-in-one lentiviral vector for simultaneous expression of two sgRNAs, Cas9 and eGFP regulated by the U6, H1, and EFS promoters. **c**, Genomic PCR analysis of edited and control A673 cells using oligonucleotides flanking the DNA targeted by sgE3 and sgF8 (*n* = 3, independent studies). The 300 bp PCR product denotes deletion of the DNA fragment between the loci targeted by sgE3 and sgF8. PCR analysis was performed on extracted DNA of cells at day 2, 4, and 6 post-transduction (pt). *ALBUMIN* was used as an internal control of the PCR reaction. Bottom panel shows a representative Sanger sequencing chromatogram of the PCR products. **d** RT-PCR products from edited and control A673 Ewing sarcoma cells (*n* = 3, independent studies). Analysis was done using extracted RNA of cells at day 2, 4, and 6 pt. Arrows depict the sizes of wild-type (961 bp) and deleted (150 bp) RT-PCR products. *GAPDH* was used as an internal control of the RT-PCR reaction. Bottom panel shows a representative Sanger sequencing chromatogram of RT-PCR products. **e** Western blotting of EWSR1-FLI1 in A673 cells. Analysis was done using total protein extracted from cells at day 3, 6, and 10 post-transfection using an antibody specific for FLI1. GAPDH was used as an internal control of the assay. LTR: Long term repeat; P2A: porcine teschovirus-1 2A self-cleaving peptide; WPRE: Virus (WHP) posttranscriptional regulatory elements.

significantly blunted clonogenic capacity (51%, 62%, 49%, and 56%, respectively) irrespective of the sgRNA pair used (Supplementary Fig. 2b). Cleavage with sgE3-sgF8 resulted in high deletion efficiency, generating the largest (27.7 kb) EF deletion and resulting in the complete loss of the EWSR1 transactivation domain and a frameshift alteration of the entire FLI1 DNA-binding region (Supplementary Fig. 3a). Accordingly, this combination was chosen for further study. Notably, targeted deep-sequencing of sgE3- or sgF8-targeted A673 cells revealed 62% and 66% insertion/deletion (indels) in EWSR1 and FLI1 on-target sites, respectively (Supplementary Table 2a, b). For subsequent targeting of EF, sgE3 and sgF8 were cloned into an all-in-one expression plasmid[34] (pLV-U6sgE3-H1sgF8-EFSCas9-2A-eGFP; hereafter termed LVCas9_EF) expressing sgE3 and sgF8 from the U6 and H1 RNApol III promoters, respectively, together with the simultaneous expression of Cas9 and GFP proteins separated by a 2A self-cleaving peptide (Fig. 1b).

**CRISPR-mediated deletion selectively reduces FO products.** We first examined the capacity of LVCas9_EF to generate EF deletions in A673 and also RD-ES Ewing sarcoma cells, which harbor different EF isoforms. The pLV-U6sgNT-EFSCas9)[33] single non-targeting vector (hereafter termed LVCas9_NT) was used as a control (Supplementary Table 1). PCR analysis of the genomic regions spanning the intronic cleavage sites

(Supplementary Table 1) revealed a PCR band verified by Sanger sequencing both in A673 cells (Fig. 1c) and in RD-ES cells (Supplementary Fig. 2c). Consistently, RT-PCR and western blot analysis confirmed that the simultaneous expression of sgE3 and sgF8 induced a robust loss of EF mRNA and protein, respectively (Fig. 1d, e and Supplementary Fig. 2d). We designed a quantitative qRT-PCR-based expression assay for FO to estimate rates of deletion, and found a significant 4.5-fold decrease in FO expression in edited A673 cells at day 3 post-transduction (p.t.) (Supplementary Fig. 2e). Deep sequencing of the deletion PCR amplicons indicated a preference (~60%) for precise ligation of cut products (Supplementary Fig. 2f). Finally, targeted deep-sequencing analysis revealed that EF-targeted human primary mesenchymal stem cells (hMSC) cells perform accurate splicing of EWSR1 exons. To rule out any possible bias in the use of a non-targeting control, we also tested a second control plasmid (LVCas9_dC1/2) targeting two unrelated genomic loci that generate two genomic DSB in two different chromosomes. Results showed no changes to cell viability in transduced A673 cells (Supplementary Table 1 and Supplementary Fig. 2g).

**CRISPR FO targeting does not impair genomic stability.** We next evaluated whether our CRISPR-mediated *EF* targeting strategy induces collateral genomic alterations. We used as wild-type control clinically-relevant healthy hMSC, which are

considered the cell-of-origin for human sarcomas[35]. Sequencing analysis of *EF*-targeted hMSCs confirmed the presence of a repertoire of indels (~55% and 68%, *EWSR1* and *FLI1*, respectively) at the targeted regions (Supplementary Fig. 4a). To assess possible side-effects that could affect proliferation of healthy cells, we treated hMSCs with LVCas9_NT or LVCas9_EF, finding no difference in proliferation rates in long-term culture (Supplementary Fig. 4b). Also, hMSC transduced with LVCas9_EF did not produce tumors when injected into immunosuppressed mice. G-banding karyotype analysis 7 days after transduction revealed no numerical or structural abnormalities in hMSC, irrespective of the vector used (Supplementary Fig. 4c), and high-density genome-wide array comparative genomic hybridization (aCGH) analysis confirmed no large copy number deletions across the genome (Supplementary Fig. 4d). hMSCs were also negative for *EWSR1-FLI1* translocation products by FISH and PCR analysis at 15- and 30-days p.t. (Supplementary Fig. 4e, f and Table 1). Finally, qRT-PCR analysis of edited hMSCs confirmed non-significant variation in the levels of *EWSR1* and *FLI1* expression (Supplementary Fig. 4g). To test for potential off-target cleavage activity, we performed next-generation sequencing (NGS) of PCR amplicons spanning the 50 more probable in silico-predicted off-target sites of sgE3 and sgF8. The threshold percentage for off-target activity was set at 0.10%, since this was the highest percentage of indel-containing reads in control samples, considering as noise indel-containing reads below this percentage[36]. For every off-target site analyzed, indel-containing reads represented less than 0.10% of the total, with the exception of one predicted site (Supplementary Table 3, Supplementary Fig. 4h). Although this site showed indel-containing reads at a slightly higher frequency (~0.12%) than the threshold, the sequence variations found were also present in both controls and edited samples, suggesting that they were not bona fide Cas9 off-targets. Thus, targeted NGS ruled out mutations in *EF*-targeted hMSCs, A673, and RD-ES cells harvested 7 days after LVCas9_EF transduction. Overall, these data show that the therapeutic targeting of the *EF* FO seems highly specific and does not interfere with non-rearranged alleles in human control healthy cells.

**Targeting of a driver FO inhibits tumor cell growth in vitro.** We utilized in vitro assays to examine the functional consequences of targeting *EF*. Transduction with LVCas9_EF, but not with LVCas9_NT, resulted in a 65.5% and 93.9% decrease in A673 and RD-ES cell growth, respectively (Fig. 2a, b), and a 60.5% and 70.3% decrease, respectively, in clonogenic capacity (Fig. 2d, e). No changes were found in LVCas9_EF-transduced control U2OS cells (non-*EWSR1-FLI1*) (Fig. 2c, f). Consistent with the growth phenotype observations, *EF*-targeted cells had higher levels (5–8-fold) of apoptosis, measured both by propidium iodide staining (subG1 cells, Fig. 2g) and caspase-3 analysis (Fig. 2h), than control transduced cells, suggesting that CRISPR-mediated abrogation of *EF* is sufficient to inhibit the survival, proliferation, and clonogenicity of *EF*-expressing Ewing sarcoma cells in vitro. As the induction of two DSBs in the same chromosome has been related to inversion events[37], we checked for the presence of this type of rearrangement in A673 cells. PCR analysis revealed the presence of inversion products at 3 days p.t.; however, inversion carrier cells were lost during cell culture and no inversion products were detectable at day 21 p.t. (Supplementary Fig. 2h). In silico analysis predicted that an inversion of the FO targeted region will produce a truncated protein similar to that induced by DNA deletion (Supplementary Fig. 3a), which is compatible with a similar mechanism of cell death of inversion carrier cells.

**FO deletion controls tumor growth in a xenograft model.** We chose adenovirus (AdV) vectors to test the in vivo applications of the targeting approach because of their high gene transfer efficiency, their ability to infect cancer cells, their capacity for large DNA transgenes (allowing co-delivery of sgRNAs and Cas9 within single vector particles) and also because they do not readily integrate into host genomes[38]. To test whether AdV-based in vivo delivery of the FO-targeting CRISPR machinery can control tumor growth, we subcutaneously injected A673 cells into immunodeficient mice and allowed tumors to develop for 10 days (~150 mm$^3$ in size) (Fig. 3a). Tumors were then injected with 100 μL of PBS containing $2.5 \times 10^9$ plaque-forming units (pfu) of the AdCas9_EF vector containing the same cassette as described above for the LV vector (sgE3 and sgF8 together with Cas9 and GFP proteins), or the equivalent non-targeting AdCas9_NT control vector, or PBS, at day 10, 13, 16 and 19 (Fig. 3a). Results revealed that compared with AdCas9_NT- or PBS-treated mice, tumor size in AdCas9_EF-treated mice was reduced by ~70% at sacrifice ($1345.2 \pm 685.2$ and $1144.0 \pm 213.1$ vs $375.7 \pm 114.9$ mm$^3$, respectively, $p < 0.01$) (Fig. 3b, c). Immunohistochemistry analysis showed a robust decrease in the number of Ki67+ cells in *EF*-targeted tumor sections (90% and 88% vs 12%, $p < 0.001$) and an increase in the number of apoptotic cells (9% and 10% vs 45%, $p < 0.001$) (Fig. 3d, e). Of note, 40% of xenografted mice treated with AdCas9_EF survived beyond day 80, whereas all PBS- or AdCas9_NT-treated mice succumbed to the disease by day 40 (Fig. 3f). Hematoxylin and eosin (H&E) staining of tumors revealed a significantly lower number of viable tumor cells and more extensive necrotic regions in *EF*-targeted tumors (Supplementary Fig. 5a). Immunohistochemistry for Cas9 confirmed adenoviral delivery into the tumors (Fig. 3d). Immunohistochemistry analysis of the tumors from AdCas9_EF-treated mice showed the colocalization of adenovirus expression (Cas9 immunostaining) with high and low expression of caspase-3 and Ki67, respectively (Fig. 3g). Because we used the Hds:Athymic Nude-Foxn1nu mouse strain, which lacks T cells and shows a partial defect in B cell development (with a weakened innate immune system), we tested for the presence of tumor-infiltrating CD45 + leukocyte cells. Results showed that the necrotic region was surrounded by tumor-infiltrating murine CD45 + leukocytes (Fig. 3g). Targeted deep-sequencing analysis of the tumors at sacrifice revealed that *EF*-targeted xenografted tumors analyzed 6 weeks after AdV injection were composed of non-edited (85%) or partially-edited (one locus; 12% of *EWSR1* or 3% of *FLI1*) cancer cells (Supplementary Table 2e, f). No cells harboring an *EF* deletion were present in the tumors suggesting that CRISPR-mediated abrogation of *EF* is sufficient to inhibit survival of Ewing sarcoma cells in vivo. Furthermore, targeted deep-sequencing of PCR amplicons covering the 50 most probable predicted off-target sites associated with sgE3 and sgF8 ruled out mutations in *EF*-targeted xenografted tumors analyzed 6 weeks after AdV injection (Supplementary Table 3).

**FO deletion controls tumor growth in PDX models.** To further investigate the utility of the approach, we used three patient-derived xenograft (PDX) models of Ewing sarcoma. The models were established by subcutaneous implantation of Ewing sarcoma PDXs into immunodeficient mice, which were allowed to develop for 12 days (tumors ~150 mm$^3$ in size) (Fig. 4a). FISH analysis showed the presence of the t(11;22) translocation (Supplementary Fig. 5b). We next examined the capacity of FO targeted deletions to reduce PDX growth by measuring tumor size. As described earlier, *EWSR1-FLI1* targeted therapy was administered by adenoviral delivery using $2.5 \times 10^9$ pfu of the AdCas9_EF in 100 μL of PBS (or a PBS control) at day 12, 14, 16 and 19 (Fig. 4a).

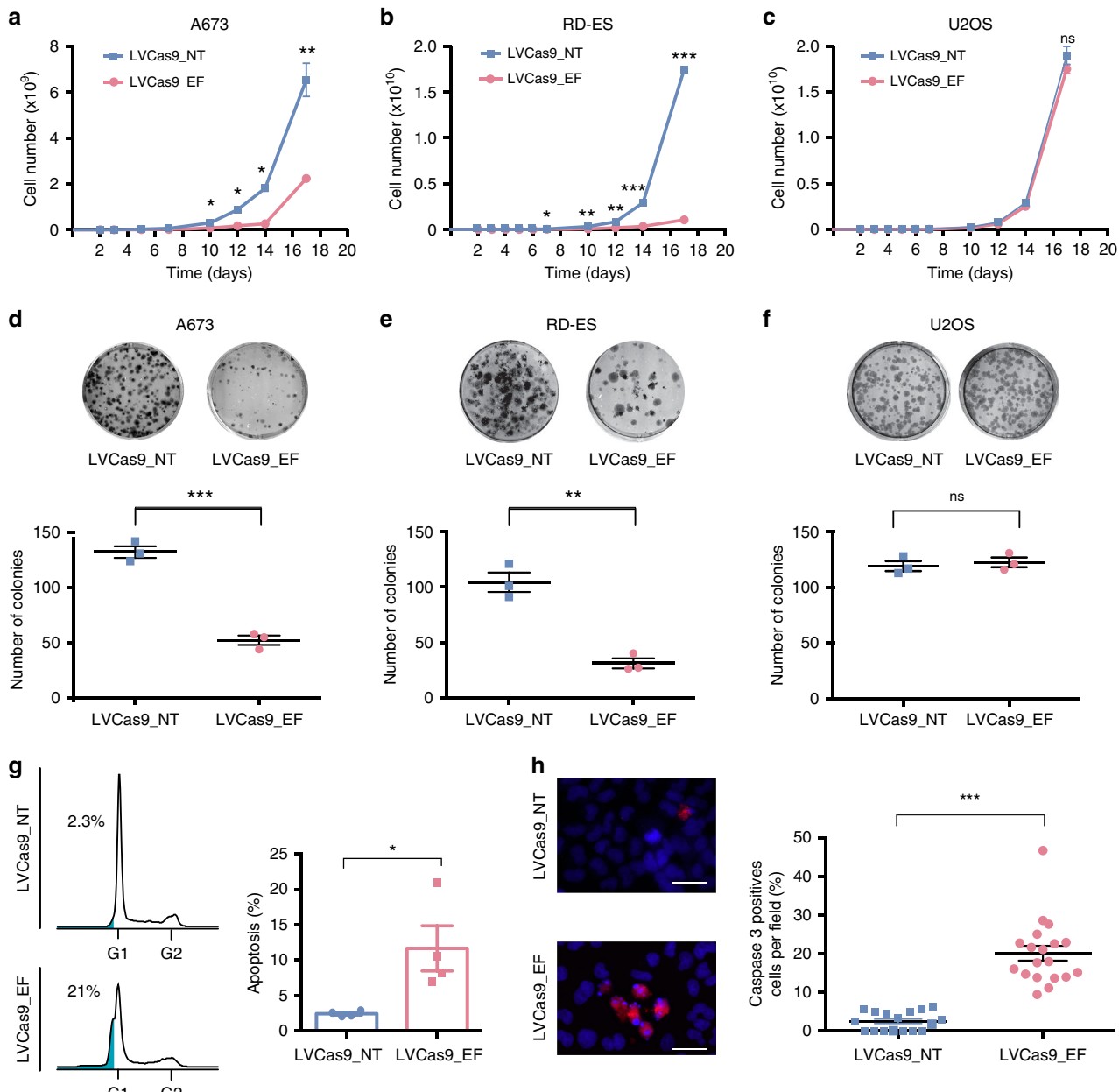

**Fig. 2 CRISPR-mediated targeting of *EWSR1-FLI1* inhibits in vitro cell growth. a–c** Growth rate assay curves of A673 (**$p = 0.01$), RD-ES (***$p = 0.0005$) and U2OS edited (LVCas9_EF) and control (LVCas9_NT) cells. ($n = 6$ independent experiments). **d–f** Representative crystal violet staining of A673 (***$p = 0.0003$), RD-ES (**$p = 0.0017$) and U2OS experimental and control cells. Bottom panels show the statistical analysis of the number of colonies. ($n = 3$ independent experiments). **g** Left panel shows DNA profile analysis by propidium iodide staining and flow cytometry. FlowJo v10.0.7 flow cytometry analysis of DNA fragmentation by subG1 population (cells with fragmented DNA) quantification. The percentage of cellular apoptosis was calculated using the percentage of the subG1 peak. Right panel shows graphical representation of subG1 analysis. ($n = 4$ independent experiments) *$p = 0.027$. **h** Representative images of caspase-3-immunostained A673 cells. Scale bars, 50 μm. Right panel shows the percentage of caspase-3-positive cells per field analyzed. ($n = 520$ (LVCas9_NT) and $n = 613$ (LVCas9-EF) cells examined over three independent experiments), ***$p = 1.07$e-9. The error bars indicate the s.e.m. for the averages across the multiple experiments; p-values are represented (ns non-significant; *$p \leq 0.05$; **$p \leq 0.01$; ***$p \leq 0.001$). Statistical are two-tailed unpaired *t*-test.

Results showed that tumors were >74% smaller in AdCas9_EF-treated mice at sacrifice than in PBS-treated mice (334.87 ± 28.01 vs 1321.70 ± 249.77 mm³, $p < 0.001$) (Fig. 4b, c). Immunohistochemistry analysis of tumor sections showed significantly fewer Ki67+ cells in *EF*-targeted tumors (18% vs 73%, $p < 0.001$) and an increased number of apoptotic cells (57% vs 1.5%, $p < 0.001$) (Fig. 4d, e). H&E staining also revealed a significantly lower number of viable tumor cells and more extensive necrotic regions in *EF*-targeted tumors (Fig. 4d). Immunohistochemistry for Cas9

confirmed adenoviral delivery into the tumors (Fig. 4d). Of note, 100% of PDX mice treated with AdCas9_EF survived beyond day 52% and 50% beyond day 70, whereas all PBS control mice succumbed to the disease by day 40 (Fig. 4f). Taken together, these results show that in vivo adenoviral delivery of *EF*-targeting CRISPR components controls sarcoma growth.

**Combined deletion and chemotherapy enhances tumor regression**. We next investigated the possible synergy between

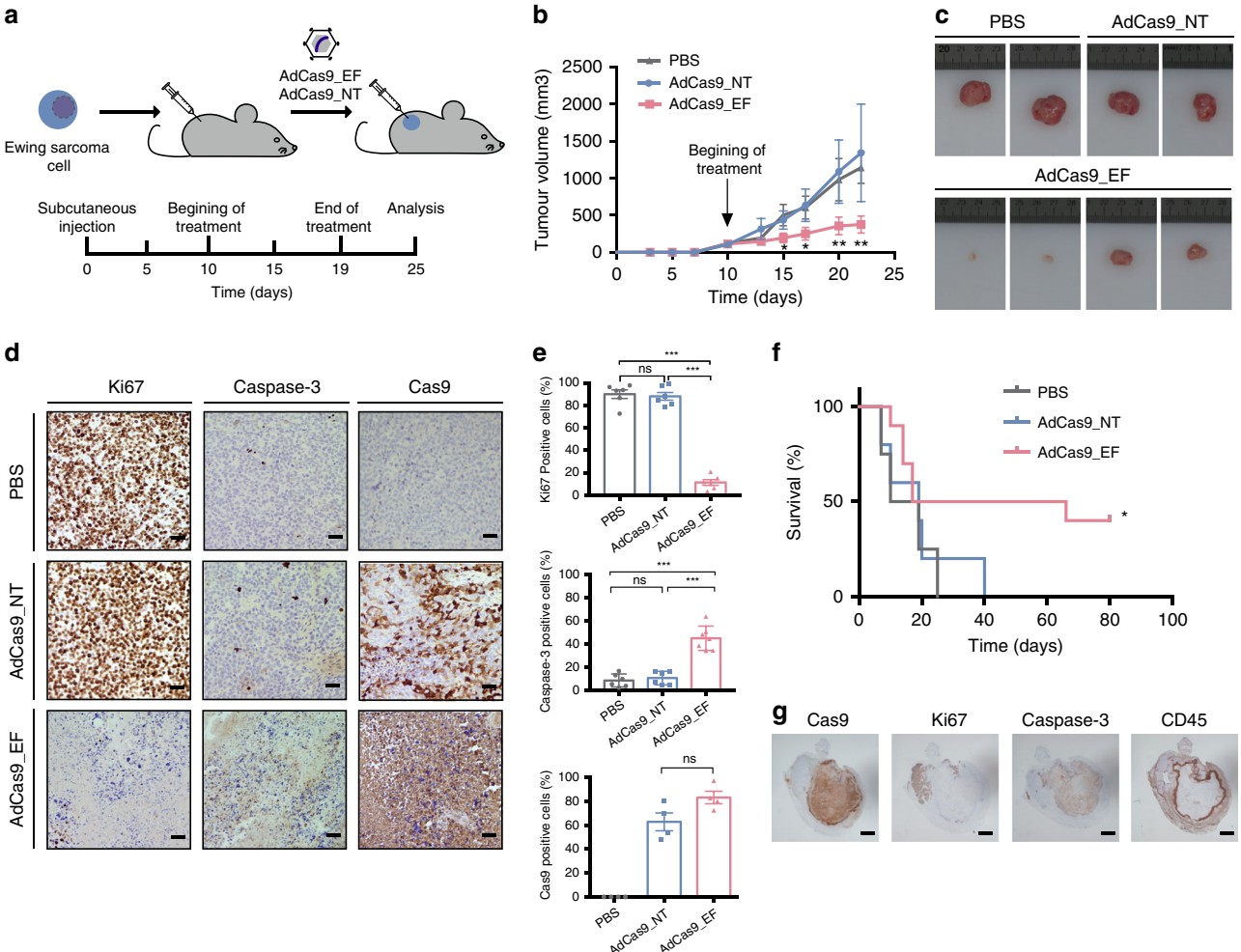

**Fig. 3 Deletion of *EWSR1-FLI1* inhibits tumor growth in xenografted models. a** Diagram showing the approach for the in vivo xenograft treatment. A673 cells were subcutaneously injected into immunodeficient mice and tumors were allowed to develop (~150 mm³ in size). Tumors were then injected with AdCas9_EF edition vector, AdCas9_NT control vector or PBS at days 10, 13, 16, and 19. Mice were sacrificed when tumor volume reached 1500 mm³. **b** Tumor growth over 23 days of analysis. (PBS *n* = 6; AdCas9_NT *n* = 6; AdCas9_EF *n* = 15 animals), **p = 0.004. **c** Images of representative tumors. **d** Representative immunostaining for Ki-67, caspase-3 and Cas9 in A673 experimental and control tumors (*n* = 3 independent samples). Scale bars, 50 µm. **e** Statistical analysis of the percentage of Ki-67- (PBS vs AdCas9_EF ***p = 1e-8, AdCas9_NT vs AdCas9_EF ***p = 6e-9), caspase-3- (PBS vs AdCas9_EF ***p = 1e-5, AdCas9_NT vs AdCas9_EF ***p = 5e-5) and Cas9-positive cells. **f** Kaplan–Meier survival curve comparing mice treated with experimental and control AdV or PBS. (PBS *n* = 4; AdCas9_NT *n* = 5; AdCas9_EF *n* = 10 animals), *p = 0.046. **g** Representative immunostaining for Cas9, Ki-67, caspase-3, and CD45 in an AdCas9_EF-treated xenografted tumor. The error bars indicate the s.e.m. for the averages across the multiple experiments; *p*-values are represented (ns non-significant; *p ≤ 0.05; **p ≤ 0.01; ***p ≤ 0.001). A two-tailed unpaired *t*-test was used for statistical analysis of **b**, **d**, and log-rank test for **f**.

our genomic editing approach to efficiently eliminate cancer cells and a standard-of-care chemotherapeutic option for Ewing sarcoma with doxorubicin. We examined the additive effects of combined doxorubicin and *EF* deletion both in vitro and in vivo. Corresponding FO deletion and treatment of A673 cells with 0.05 µg/ml doxorubicin resulted in a greater reduction of cell viability (50%) than single treatments (39% or 42% with AdCas9_EF or doxorubicin treatment alone, respectively) (Fig. 5a). For in vivo analysis, we utilized the xenograft models generated as described earlier, divided into four treatment groups. Mice were treated with 1.5 mg/kg doxorubicin (intraperitoneal injection, weekly, for two weeks) or with 2.5 × 10⁹ pfu of the AdCas9_EF or AdCas9_NT virus in PBS (intratumour injection, at day 9, 12, 15, and 18) (Fig. 5b). Three groups received monotherapy of AdCas9_NT, doxorubicin or AdCas9_EF, while the remaining group was treated with combined therapy (doxorubicin + AdCas9_EF), and animals were monitored for 80 days. Results

showed that mice treated with combined therapy showed a greater reduction in tumor size at sacrifice (92.12%, 115.28 ± 28.35 mm³) than mice treated with monotherapy (83.7%, 237.4 ± 47.39 mm³ AdCas9_EF or 70.4%, 433.05 ± 56.91 mm³ doxorubicin) as compared with control mice (1463.82 ± 278.17 mm³) (Fig. 5c). These results indicate that combined therapy is more effective than monotherapy. Of note, 66% of xenografted mice treated with combined therapy survived beyond day 80 compared with 33% of AdCas9_EF-treated mice. By contrast, all the mice that received doxorubicin monotherapy succumbed to the disease by day 64 (Fig. 5d).

**Validation as a method for elimination of FO-addicted cells.**
To validate the editing strategy as a universal strategy for elimination of FO-driven cancer cells, we applied the same approach in a cellular model harboring *BCR-ABL1* (*BA*), a classical tyrosine kinase FO hallmark of chronic myeloid leukemia (CML).

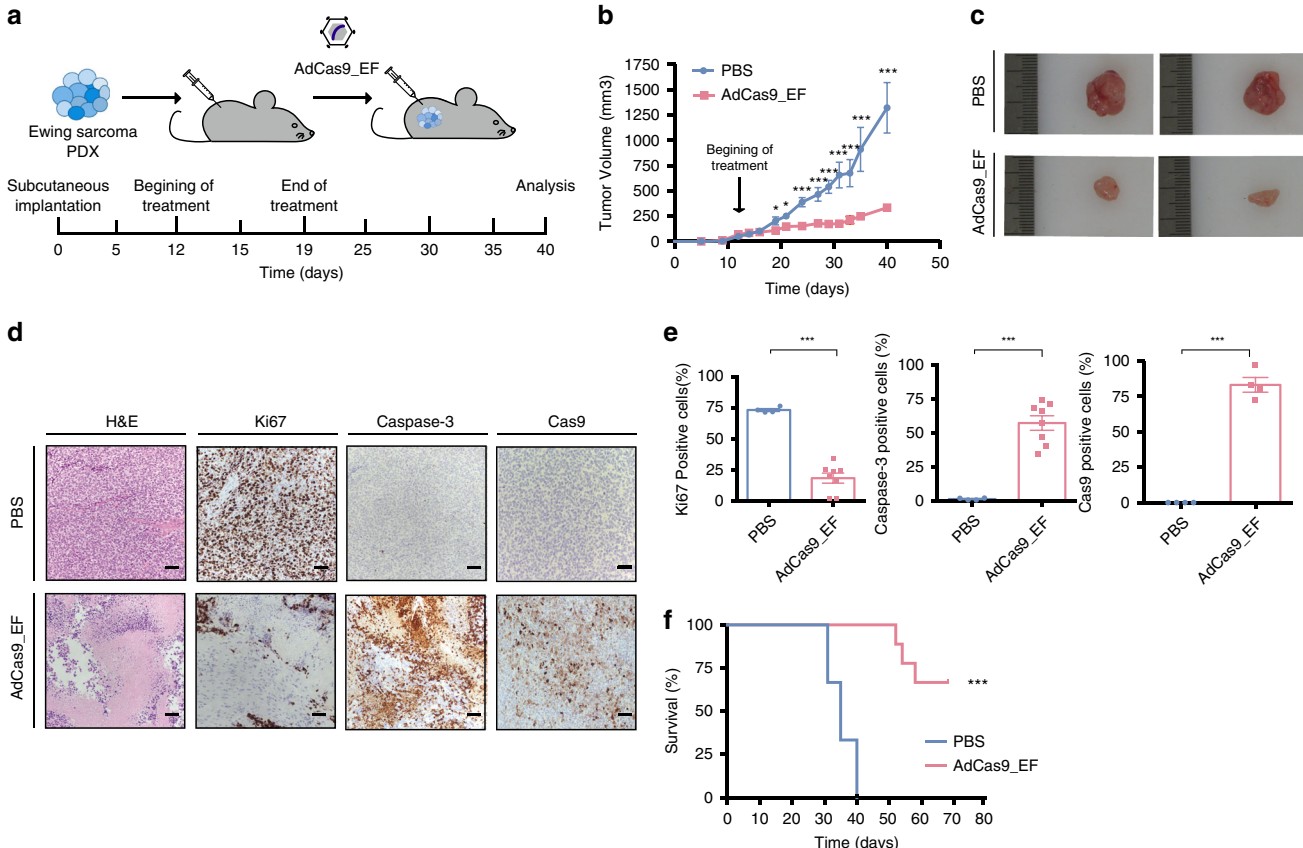

**Fig. 4 Deletion of *EWSR1-FLI1* inhibits tumor growth in PDX models. a** Diagram showing the approach for the AdV-based in vivo treatment. PDXs were implanted into immunosuppressed mice and tumors were allowed to develop (~150 mm³ in size). Tumors were then injected with AdCas9_EF edition vector or PBS control vector at days 12, 14, 16, and 19. Mice were sacrificed when tumor volume reached 1500 mm³. **b** Tumor growth over the 40 days following subcutaneous PDX implantation. (PBS $n = 3$; AdCas9_sgEF $n = 9$ animals) (***$p = 3e-5$). **c** Representative tumors of control and experimental mice sacrificed 40 days after PDX implantation. **d** Representative images of histological H&E staining, Ki-67, caspase-3, and Cas9 immunostaining of A673 experimental and control cells. Scale bars, 500 μm (4×) or 50 μm (20×). **e** Percentage of Ki-67-(***$p = 3e-6$), caspase-3-(***$p = 3e-5$) and Cas9-positive cells per field analyzed (***$p = 4e-6$). ($n = 3$ independent samples). **f** Kaplan–Meier survival curve comparing mice treated with experimental and control AdV. (PBS $n = 3$; AdCas9_sgEF $n = 9$ animals), ***$p = 0.0002$. Plot shows medians and ranges; error bars indicate the s.e.m. for the averages across the multiple experiments; $p$-values are represented (*$p \leq 0.05$, **$p \leq 0.01$, ***$p \leq 0.001$). A two-tailed unpaired $t$-test was used for statistical analysis of **b**, **e**, and log-rank test for **f**.

Four sgRNAs targeting *BCR* intron 8 and *ABL* intron 1 regions were designed to induce a 133.9 kb deletion of the *BA* FO (Supplementary Table 1), resulting in an extensive deletion of the *BCR* transactivation domain and a frameshift alteration of the entire *ABL1* DNA-binding domain (Fig. 6a and Supplementary Fig. 3b). Four sgRNA combinations (sgB8.1-sgA1.1, sgB8.1-sgA1.2, sgB8.2sgA1.1, and sgB8.2-sgA1.2) were cloned into the pLV-U6<sup>B</sup>H1<sup>A</sup>-C9G vector as before, and were electroporated into the *BA*-expressing CML cell line K562 (Fig. 6b). Electroporation was chosen as the delivery method because of the low transduction efficiency of lentiviral vectors in K562 cells[39]. All four sgRNA combinations efficiently abrogated the expression of *BA* as confirmed by RT-PCR and Sanger sequencing (Fig. 6c). For any given sgRNA combination, electroporated K562 cells showed a significant decrease (~85%) in clonogenic capacity in vitro (Fig. 6d) concomitant with an increase in apoptosis (3% vs 15%, $p < 0.001$) relative to equivalent cells electroporated with a nontargeting vector (Supplementary Fig. 6a). The use of a control plasmid (LVCas9_dC3/4) targeting two unrelated genomic loci in two different chromosomes ruled out any effects on cell viability in K562-treated cells due to the induction of two DSBs (Supplementary Table 1 and Supplementary Fig. 6b, c). Because cleavage with sgB8.1-sgA1.1 resulted in the highest reduction in

colony formation and the highest percentage of apoptosis rate this combination was chosen for further study. For subsequent targeting of *BA*, sgB8.1 and sgA1.1 were cloned into an all-in-one expression plasmid[34] (pLVCas9_BA). Targeted deep-sequencing of sgB8.1 or sgA1.1-targeted K562 cells revealed 85% and 82% indels in *BCR* and *ABL1* on-target sites, respectively (Supplementary Table 2). Similarly, targeted deep-sequencing of PCR amplicons covering the most probable predicted off-target sites associated with sgB8.1 and sgA1.1 ruled out mutations in *BA*-targeted K562 cells after pLVCas9_BA electroporation (Supplementary Table 3). qRT-PCR analysis of FO expression at day 2 post-electroporation (p.e.) of pLVCas9_BA edited K562 cells used to estimate rates of deletion revealed a significant 5.0-fold decrease in FO expression (Fig. 6e). We also analyzed the presence of inversions in K562 cells as before. PCR analysis revealed the presence of inversion products at 2 days p.e.; however, inversion carrier cells were lost during cell culture and no inversion products were detectable at day 7 p.e. (Supplementary Fig. 6d). In silico analysis showed that an inversion of the FO targeted region will produce a truncated protein similar to the one induced by DNA deletion (Supplementary Fig. 3b). We used cord blood-derived human hematopoietic progenitor hCD34+ cells as a control cell of origin for CML. In vitro analysis of targeted

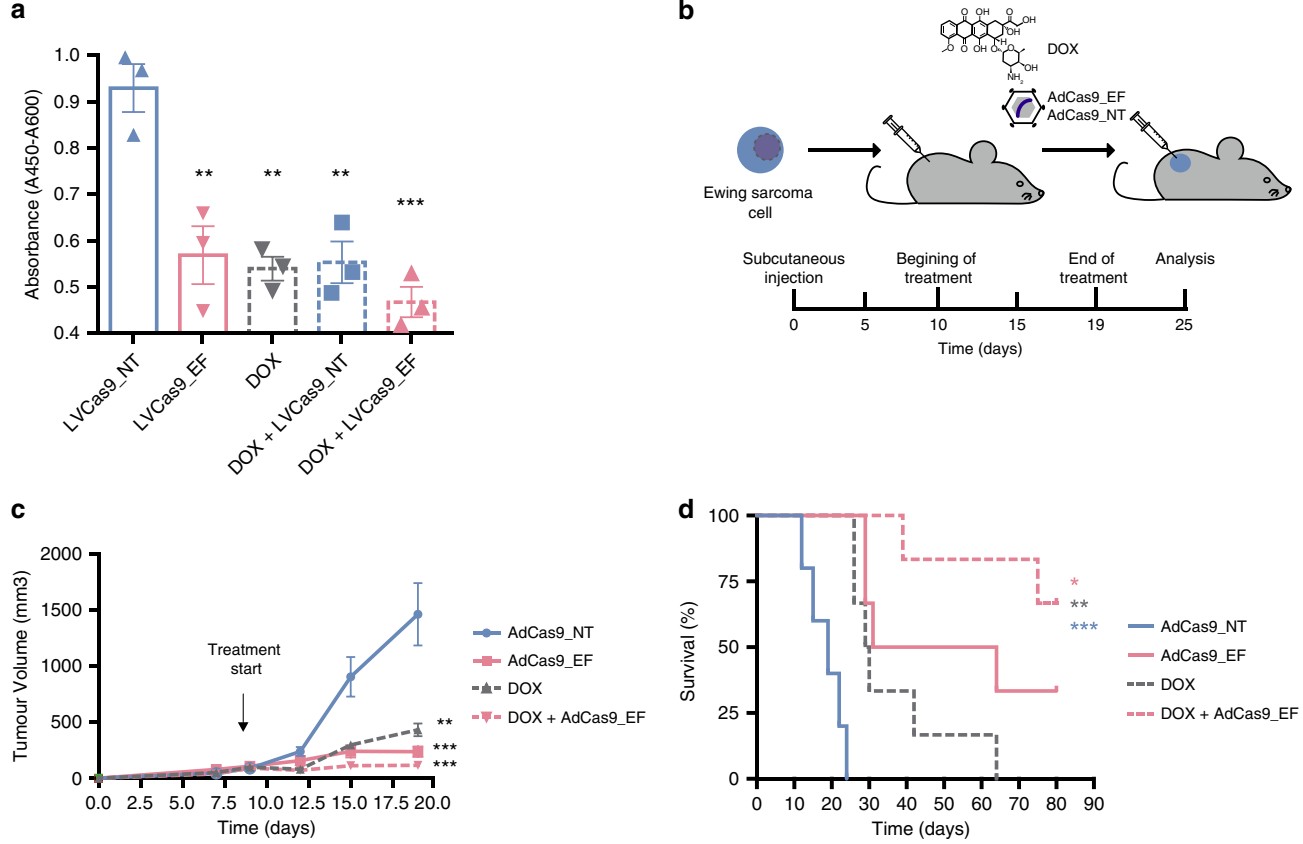

**Fig. 5 Combined approach of *EWSR1-FLI1* deletion with doxorubicin inhibits tumor growth in xenograft models. a** WST-1 cell proliferation analysis of A673 cell treated with CRISPR-deletion or doxorubicin as monotherapy, or as combined therapy and controls ($n = 3$ independent experiments) (LVCas9_NT vs LVCas9_EF **$p = 0.01$, LVCas9_NT vs Dox **$p = 0.002$, LVCas9_NT vs Dox + LVCas9_NT **$p = 0.005$, LVCas9_NT vs Dox + LVCas9_EF ***$p = 0.0001$). **b** Diagram showing the approach for the in vivo xenograft treatment. A673 cells were subcutaneously injected into immunodeficient mice and tumors were allowed to develop (~150 mm$^3$ in size). Tumors were then injected with AdCas9_EF edition vector, doxorubicin (DOX), AdCas9_NT control vector or their combination at days 9, 12, 15, and 18. Mice were sacrificed when tumor volume reached 1500 mm$^3$. **c** Tumor growth over the 19 days following subcutaneous cells implantation. (AdCas9_NT $n = 5$; AdCas9_sgEF $n = 6$; DOX $n = 6$, DOX + AdCas9_EF $n = 6$ animals) (AdCas9_NT vs DOX **$p = 0.003$, AdCas9_NT vs AdCas9_EF ***$p = 9e-4$, AdCas9_NT vs DOX + AdCas9_EF ***$p = 5e-4$). **d** Kaplan–Meier survival curve comparing mice treated with CRISPR-deletion or DOX as monotherapy, or as combined therapy and controls. (AdCas9_NT $n = 5$; AdCas9_sgEF $n = 6$; DOX $n = 6$, DOX + AdCas9_EF $n = 6$ animals) (DOX + AdCas9_EF vs AdCas9_EF *$p = 0.035$, DOX + AdCas9_EF vs DOX **$p = 0.003$, DOX + AdCas9_EF vs AdCas9_NT ***$p = 0.0007$). Plot shows medians and ranges; error bars indicate the s.e.m. for the averages across the multiple experiments; $p$-values are represented (*$p \leq 0.05$, **$p \leq 0.01$, ***$p \leq 0.001$). A two-tailed unpaired $t$-test was used for statistical analysis of **a**, **c**, and Log-rank test for **d**.

CD34 + cells revealed no difference in proliferation in long-term culture, suggesting no production of collateral cancer-driven genomic alterations (Supplementary Fig. 6e). No *BCR-ABL1* translocation products were detected in cells by PCR or FISH assays conducted at 15- and 30-days p.e. (Supplementary Fig. 6f, g, Supplementary Table 1). We then evaluated whether AdV delivery of the *BA*-targeting CRISPR machinery (sgB8.1-sgA1.1) also controls the growth of *BA*-driven tumors in vivo. Accordingly, K562 cells were subcutaneously injected into immunodeficient mice and tumors were allowed to develop for 15 days (~150 mm$^3$ in size) (Fig. 6f). Tumors were then injected with $2.5 \times 10^9$ pfu of the AdCas9_BA editing vector or the AdCas9_NT control vector at day 15, 18, and 21. Adenoviral delivery of AdCas9_BA profoundly inhibited tumor growth, resulting in an 88% decrease in tumor size at sacrifice ($281 \pm 63$ vs $1152 \pm 158$ mm$^3$ ($p < 0.001$) compared with control tumors (Fig. 6g, h). Strikingly, 50% and 22% of xenografted mice treated with AdCas9_BA were alive after 38 and 64 days, respectively, whereas all AdCas9_NT-treated mice succumbed

to the disease by day 34 (Fig. 6i). As observed in the Ewing sarcoma model, the efficiency of the AdCas9_BA editing vector was also associated with a robust decrease in the number of Ki67+ cells (82% vs 17%, $p < 0.001$) and an increase in apoptotic cells (2.5% vs 10%, $p < 0.05$) (Supplementary Fig. 6h, i). H&E staining also revealed a significantly lower number of viable tumor cells and more extensive necrotic regions in *BA*-targeted tumors (Supplementary Fig. 6h). A combinatorial approach using *BA* deletion with a standard-of-care therapeutic option in CML (imatinib, 0.105 μg/ml) resulted in a significant reduction in K562 cell viability compared with any of the treatments in monotherapy (Fig. 6j). Targeted deep-sequencing analysis of the tumors at sacrifice revealed that *BA*-targeted xenografted tumors analyzed 6 weeks after AdV injection were composed of non-edited (94%) or partially-edited (one locus; 3% of *BCR* or *ABL1*) cancer cells (Supplementary Table 2g, h). Taken together, these results confirm the efficacy of CRISPR-mediated targeting of sarcoma (*EF*)- and leukemia (*BA*)-associated FO for cancer cell-directed elimination.

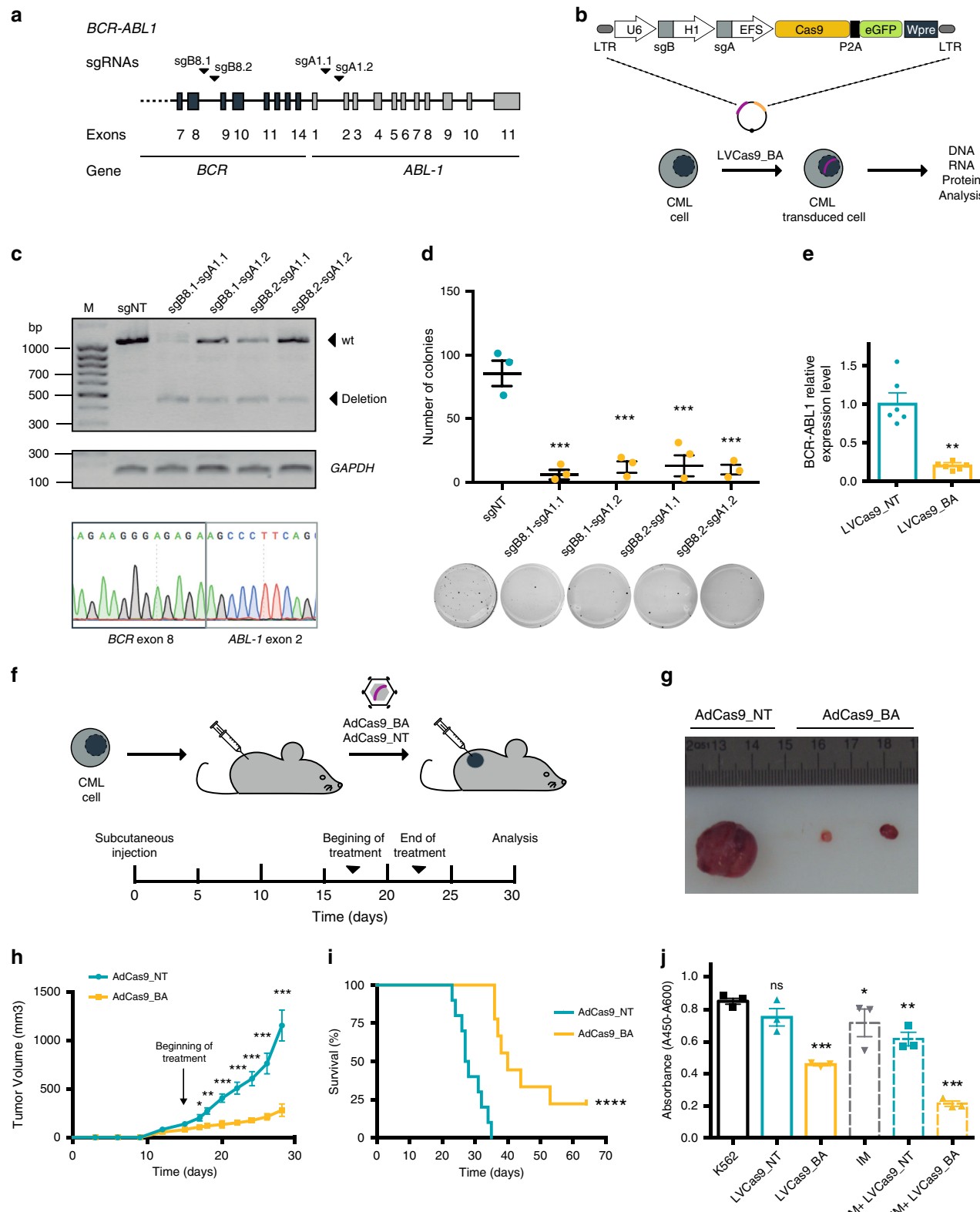

## Discussion

FOs are ideal therapeutic targets for the development of new directed cancer treatments, owing to their cancer-driving roles, their restriction to cancer cells and the reliance of tumors on them. Unfortunately, FOs are challenging to target directly with candidate drugs, and although some successful anti-cancer drugs have been developed based on the ability to target FOs[40,41], the development of new therapies are still needed. The ability to precisely manipulate cancer cell genomes to correct or eliminate cancer-causing aberrations by highly-efficient CRISPR/Cas9 genome editing opens new possibilities to develop FO-targeted options to eliminate cancer cells. In the present study, we describe a simple and efficient genome editing strategy specifically targeting FOs in cancer cells. Our CRISPR/Cas9-based approach

**Fig. 6 Strategy validation in CML-initiating *BCR-ABL* model. a** Schematic representation of the *BCR-ABL1* locus showing the sgRNAs targeting introns 8 of *BCR* and 1 of *ABL1*. **b** Schematic illustration of the vector and the approach for in vitro treatment. **c** Agarose gel electrophoresis of BCR-ABL RT-PCR products obtained from experimental and control K562 cells electroporated with four combinations of sgRNAs. RT-PCR analysis was done using RNA extracted from cells at day 2 post-nucleofection. Arrows depict the sizes of wild-type (1125 bp) and deleted (458 bp) RT-PCR products. *GAPDH* was used as an internal control of the RT-PCR reaction. Bottom panel shows a representative chromatogram of Sanger sequencing analysis of the RT-PCR products. **d**, Representative images and graphical representation of colony formation of K562 experimental and control cells (stained with nitrotetrazolium blue). ($n = 3$ independent experiments). sgNT vs sgB8.1-agA1.1 ***$p = $ 3e-10, sgNT vs sgB8.1-agA1.2 ***$p = $ 5e-4, sgNT vs sgB8.2-sgA1.1 ***$p = $ 3e-4, sgNT vs sgB8.2-agA1.2 ***$p = $ 1e-4. **e** *BCR1-ABL* expression level analysis. Relative expression level of *BCR-ABL1* in control (LVCas9_NT) and treated (LVCas9_EF) K562 cells measured by qRT-PCR and normalized to *GUSB* ($n = 6$ independent experiments, **$p = 0.0020$). **f** Diagram of xenograft production, adenoviral treatment and tumor growth over the 30 days following subcutaneous cell injection and four in vivo adenoviral treatments. (AdCas9_NT $n = 10$; AdCAas9_sgBA $n = 10$). **g** Representative tumors of control and experimental mice. **h** Tumor growth curve over the 28 days of study. (AdCas9_NT $n = 10$; AdCas9_sgBA $n = 10$, ***$p = $ 3e-4). **i** Kaplan–Meier survival curve comparing mice treated with experimental and control adenoviral vectors (AdCas9_NT $n = 10$; AdCas9_sgBA $n = 10$ animals, ****$p < 0.0001$). **j** WST-1 cell proliferation analysis of K562 cell treated with CRISPR-deletion or imatinib (IM) as monotherapy, or as combined therapy and controls ($n = 3$ independent experiments), K562 vs LVCas9_BA ***$p = $ 3.9e-5, K562 vs IM *$p = 0.043$, K562 vs IM + LVCas9_NT **$p = 0.006$, K562 vs IM + LVCas9_BA ***$p = $ 4e-6. Plot shows medians and ranges; error bars indicate the s.e.m. for the averages across the multiple experiments; p-values are represented (*$p \leq 0.05$, **$p \leq 0.01$, ***$p \leq 0.001$). A two-tailed unpaired *t*-test was used for statistical analysis of **d**, **e**, **h** and **j**, and Log-rank test for **i**.

induces two targeted intronic DSBs in both genes involved in a FO that, importantly, produces a cancer cell-specific genomic deletion that is dependent on the presence of the FO, and has no effect on wild-type gene expression in non-cancer cells. By targeting two established FOs, involving either a transcription factor (*EF*) or a tyrosine kinase (*BA*), we present a proof-of-concept demonstration supporting the efficacy of genome edition to eliminate cancer cells addicted to FOs. Exploiting an adenoviral in vivo delivery of CRISPR components, we further demonstrate that CRISPR/Cas9-mediated deletion in advance PDX cancer models results in efficient induction of cancer cell death and reduces tumor burden and mortality. Finally, we show that targeted CRISPR-based FO deletion combined with chemotherapy agents has an additive/synergistic effect on cell viability, tumor growth and overall survival when used in xenograft models. In the long term, this approach may provide an attractive strategy to eliminate cancer cells and we believe our findings might have applicability to other FOs beyond *EF* and *BA*. Due to the complex genomic scenario in cancer cells, including co-existence of other oncogenes, tumor suppressor genes, DNA damage response and DNA repair pathways alterations, a case-by-case FO study could shed light to the applicability of the strategy to eliminate different types of cancer cells.

A recent study has described an approach for targeting FOs using CRISPR/Cas9-based editing and the HDR DNA repair mechanism. The strategy involves directly targeting the patient-specific breakpoint to introduce a suicide gene into the genome of cancer cells[25]. Notably, this elegant approach is based on poorly efficient and breakpoint-specific design. Our system offers unique advantages over breakpoint-targeted approaches. First, it exploits the efficient NHEJ pathway active in all cell types, making the design simple and highly efficient[42], as shown here with high efficiency targeting FOs for in vivo elimination of cancer cells with significant decreases in tumor volume and increases in survival rates. Second, the strategy targets two introns flanking all breakpoint regions described in patients, making it a universal approach for all cancer cells harboring a given FO irrespective of the FO isoform or the patient-specific breakpoint. For those promiscuous FOs (such as *EWSR1*) that have been described rearranged with many different fusion partners, a combination of the original 5′ sgRNA with a set of pre-designed 3′ sgRNAs will cover all the possible FO family[31]. Third, our approach does not affect exonic sequences or the expression of wild-type alleles involved in the rearrangement.

CRISPR/Cas9-mediated genome editing could be a potential platform for cancer treatment; however, (i) targeting efficiencies,

(ii) off-target effects, and (iii) in vivo delivery remain technical limitations. As described in the present study, combinatorial approaches using gene editing with other therapeutic options (e.g., chemotherapy) further increases the efficacy of this approach, allowing better responses with lower chemotherapy doses for the elimination of cancer cells. In contrast to chemotherapy agents, which can be damaging to normal tissues, we show that the CRISPR approach is completely specific for the elimination of cancer cells[43,44]. Potential off-target events at unintended genomic sites remain important concerns that must be investigated rigorously to ensure short- and long-term safety of any gene-editing approach. We have used in vitro and in vivo assays to study the potential effect of collateral cancer-driver genomic alterations. Even though our results did not show proliferation differences, these experiments are limited, and further analysis will be needed to rule out the appearance of cancer-driven off-target modifications. Although the CRISPR-based toolbox supports diverse operations ranging from DNA and RNA editing to gene expression modulation, delivery remains a bottleneck for therapy development[45]. AdV vectors are a commonly utilized delivery vector in mouse models; they are ideal for delivery of the entire CRISPR/Cas9 system in one vector, with high transfection efficacy and they limit potential off-target effects because their genome does not integrate into the host genome[46–48]. Although the initially encountered high immunogenicity inhibited their widespread use to treat genetic disorders, the overall picture changed when it was recognized that adenoviral infection in a tumor can activate a robust immunogenic response, creating new opportunities in cancer therapy[49,50]. The number of clinical trials using oncolytic AdVs is increasing, and at present more than 32 trials are underway in different phases (https://clinicaltrials.gov)[51]. As in our study, the main delivery pathway used in these clinical trials is local administration (intratumoral injection), which shows no toxicity at specific doses and, in addition to the therapeutic effect, triggers an antitumor immune response with therapeutic effects even on not-injected distal tumors[52]. Our Hds:Athymic Nude-Foxn1 animal model characterized by partial defects in B cell development shows tumor-infiltrating CD45 + mouse leukocyte cells in the tumor. Nevertheless, additional studies on immunocompetent syngeneic mouse models could yield information about the role of AdVs as a delivery tool in cancer therapy. In the present study, we used adenoviral-based in vivo delivery of the CRISPR/Cas9 components as a proof-of-concept; however, we envision that other delivery platforms such as AAVs and non-viral (polymeric and liposome-based) vectors could be exploited to deliver the system

to tumor cells. Further development of safer and robust delivery systems and the refinement of genome editing technologies will ultimately ensure optimal targeting efficiencies and safety required for the potential clinical application of this cancer cell-directed therapy.

Overall, our data demonstrate that NHEJ CRISPR-mediated FO deletion is a selective and efficient strategy for cancer cell elimination, providing a valuable tool for basic research and could represents a promising strategy with therapeutic potential.

## Methods

**sgRNA design and generation of lentiviral constructs**. sgRNAs were designed using the online Benchling CRISPR gRNA Design tool (http://www.benchling.com). The sgRNAs chosen were based on a high specificity rank and a low potential off-target score[53]. The parental *Streptococcus pyogenes* Cas9 expression lentiviral single and double sgRNAs expression vectors, pLV-sgRNA-Cas9 and pLV-U6#1-H1#2-C9G, respectively, have been described[33,34]. sgRNAs were cloned using synthetic gBlock fragments (IDT) that were digested with PacI and XhoI (NEB) to facilitate cloning into the lentiviral backbone. When two different guides were used, the first was cloned under the control of the U6 promoter and the second was cloned under the control of the H1 promoter. The sequences for sgRNAs, gBlocks, and primers used are listed in Supplementary Table 1. All transformations were performed in *Stbl3* bacteria (Thermo Fisher Scientific).

**Cell culture**. HEK293T/17 (CRL-11268), A673 (CRL-1598), umbilical cord blood-derived hMSC (PCS-500-010), U2OS (HTB-96), and K562 (CCL-243) cells were purchased from the American Type Culture Collection (ATCC). RD-ES cells were a gift from Dr. Javier Alonso (Instituto Salud Carlos III, Madrid). HEK293T/17, U2OS, and A673 cells were maintained in Dulbecco's modified Eagle's medium (DMEM, Lonza) and K562 and RD-ES cells were maintained in Roswell Park Memorial Institute (RPMI) medium (Gibco); all were supplemented with 1% Glutamax (Life Technologies), 10 mg/ml antibiotics (penicillin and streptomycin, P/S, Gibco) and 10% fetal bovine serum (FBS, Life Technologies). hMSCs were cultured in MesenPRO RS media (Thermo Fisher Scientific) supplemented with 1% Glutamax (Life Technologies) and 10 mg/ml antibiotics (P/S, Gibco). hCD34+ were obtained freshly from the umbilical cord blood of healthy donors after informed consent was obtained and after approval by the Cord Blood Bank Transfusion Center of the Community of Madrid. Mononuclear cells were purified by gradient centrifugation (Histopaque, Sigma-Aldrich) and hCD34+ cells were purified by separation using magnetic beads (Miltenyi Biotec)[54]. hCD34+ cells were cultured in StemSpan SFEM II (StemCell Technologies) containing 100 U/mL P/S and a cytokine cocktail composed of 100 ng/mL of human stem cell factor (hSCF) and human FMS-like tyrosine kinase 3 (hFlt3L) and human thrombopoietin (hTPO) and 10 ng/mL of human interleukin 3 (hIL3) (all from Preprotech). Cells were cultured at 37 °C at 5% CO₂, 5% O₂ atmosphere in a humidified incubator. All cell lines were periodically tested for mycoplasma contamination.

**Lentivirus generation, titration, transduction**. Recombinant lentiviruses were produced by transient plasmid transfection of HEK293T/17 cells[55]. Briefly, cells were seeded at $1.1 \times 10^7$ cells/dish in 15-cm dishes the day before transfection, and were transfected by the calcium-phosphate method using 14.6 and 7.9 μg of second-generation packaging plasmids (psPAX2 and pMD.2G, Addgene #12260 and #12259, respectively) and 22.5 μg of the appropriate transfer plasmid depending on the experiment. The medium was collected after 48 h, cleared by low-speed centrifugation, and filtered through 0.45 μm PVDF filters (Millipore). Viral stocks were concentrated by ultracentrifugation in a Beckman LE ultra-centrifuge using a SW28 rotor at 20,000 × g for 2 h at 4 °C. Pellets containing lentivirus were air dried and resuspended overnight at 4 °C in 300 μL of sterile NaCl 0.9% solution (B.Braun). Viral titers were calculated by FACS analysis on transduced HEK293T cells when vectors expressed fluorescent proteins (trans-duction units/mL) and particles quantified by qPCR in supernatants (particles/mL), and were between $10^7$ and $10^8$ TU/mL. Viral aliquots were stored at −80 °C. Target cells were re-plated 24 h before transduction and transduced using a multiplicity of infection between 2 and 5. Cells were incubated at 37 °C for 6 h and then viral supernatant were replaced with fresh media.

**In vitro colony forming assays**. For A673, RD-ES and U2OS cells, colony forming assays were performed 24 h after transduction with the corresponding lentiviral vectors in six-well plates (500 cells/well)[56]. Cells were allowed to grow for 14–21 days, after which cells were fixed using methanol solution and stained with a 0.5% crystal violet solution (Sigma-Aldrich) and counted. For K562 cells, methylcellulose (StemCell Technologies) assays were performed by seeding 500 cells into 35-mm tissue culture dishes (ThermoFisher Scientific) 24 h after elec-troporation. Colonies were stained two weeks later with Nitrotetrazolium Blue chloride solution (NBT, Sigma-Aldrich) for 1 h at 37 °C and counted. All assays were performed in triplicate and repeated three times.

**Apoptotic SubG1 cell detection**. Cell cycle distribution was analyzed by mea-suring the DNA content using flow cytometry. Exponentially growing transduced cells were pelleted by centrifugation, washed with PBS (Sigma-Aldrich), resus-pended in ice-cold 70% ethanol (Sigma-Aldrich) in PBS for fixation, and were maintained at 4 °C for 24 h. Prior to analysis, fixed cells were washed with phosphate-citrate buffer (Sigma-Aldrich) and incubated with propidium iodide staining solution (Sigma-Aldrich) containing 100 μg/mL of RNase (Qiagen) for 30 min. Cellular DNA content was measured using a BD FACSCanto™ platform (BD Biosciences) and analyzed by using FlowJo® software (version 10.0.6, FlowJo).

**PCR, RT-PCR, and qRT-PCR analysis**. Genomic DNA and total RNA were extracted using standard procedures with the DNeasy Blood and Tissue Kit or the RNeasy Purification Kit, respectively (Qiagen). DNA and RNA were quantified by spectrophotometry. cDNA was generated from 1 μg of total RNA using the High-Capacity cDNA Reverse Transcription Kit (ThermoFisher Scientific). PCR and RT-PCR amplification were performed using Q5 High-Fidelity DNA Polymerase (NEB). qRT-PCR was performed in 384-well plates with 2x SYBR Green Master Mix (Applied Biosystems) using a QuantStudio 6 Detection System (Applied Biosystems). Expression levels were normalized to the housekeeping genes *ALBUMIN* for genomic DNA PCR and *GAPDH* for RT-PCR or *GUSB* for qRT-PCR. The primers used are listed in Supplementary Table 1.

**Western blotting**. Proteins were extracted using standard procedures in the presence of protease inhibitors[57]. Protein lysates were fractionated by SDS-PAGE and transferred to PVDF membranes (Millipore) using TransFi (ThermoFisher Scientific). Membranes were probed for FLI1 with a mouse monoclonal antibody (1/400; BD Pharmigen). GAPDH was used as a loading control (1/200; Abcam). Secondary antibodies were HRP-conjugated with a goat anti-mouse IgG (1/1000; Abcam) and blots were developed with the ECL reagent (GE Healthcare) and exposed to film (Kodak).

**Immunoassays**. To detect apoptosis, transduced A673 cells were seeded onto glass coverslips coated with poly-L-lysine (Cultek). After 72 h, cells were washed twice with PBS (Sigma-Aldrich), fixed in 4% paraformaldehyde (PFA; Electron Micro-scope Sci) for 12 min at room temperature (RT), permeabilized with 0.3% Triton X-100 (Sigma-Aldrich) in PBS and blocked with 3% normal goat serum (NGS; Sigma-Aldrich) in PBS for 1 h at RT. Thereafter, samples were incubated overnight at 4 °C with an anti-caspase-3 antibody (1/200; BD) diluted in PBS supplemented with 1% NGS, and then with an Alexa Fluor-594-conjugated secondary antibody (1/500; ThermoFisher Scientific) for 1 h at RT. Finally, samples were counter-stained with DAPI (Vector Labs), air dried and mounted in Vectashield mounting medium (Vector Labs). Images were acquired on a Leica DM5500B microscope (Leica Microsystems) with two lasers with excitation at 594 nm (red channel, caspase-3 detection) and 405 nm (blue channel, nuclear DAPI staining). Data were collected sequentially at a resolution of 1024 × 1024 pixels and are representative of every experiment analyzed using Cytovision v7.4 software (Leica Biosystems).

Tumor tissues were fixed with 4% PFA (Electron Microscopy Sci) in PBS overnight at 4 °C for immunohistochemistry assays[58]. Tissue sections were deparaffinized, rehydrated, blocked with 3% hydrogen peroxide (Merck) and subjected to heat-induced antigen retrieval. The following antibodies were used at the indicated dilutions: mouse anti-Ki-67 (DAKO; 1:1.000); rabbit anti-cleaved caspase-3 (1:800); rabbit anti-GFP (1:1000), mouse anti-Cas9 (1:1000) and Rabbit anti-CD45 (D3F8Q) (1:200) (all from Cell Signaling Technology). Immunostained tissue images were examined with an Olympus AX70 microscope and processed with ImageJ to estimate the labeling index (percentage of positively stained nuclear area).

**Karyotyping**. Cell cultures were arrested in metaphase by incubation with Kar-yoMAX Colcemid (Gibco; 0.1 mg/mL). Cells were treated with a hypotonic solu-tion (0.075 M KCl in ddH2O; Merck) for 30 min at 37 °C and were then fixed with Carnoy's solution (methanol and acetic acid 3:1; Sigma-Aldrich). For GTG banding, slides were trypsinized (ThermoFisher Scientific) and stained with Giemsa (ThermoFisher Scientific). Metaphase spreads were analyzed on a Zeiss Axioplan microscope with Ikaros v5.20 karyotyping platform (Metasystems GmbH). Between 40 and 100 metaphases were analyzed.

**Comparative genomic hybridization arrays**. Comparative genome hybridization analysis was performed on a dual color oligonucleotide-based array (SurePrint G3 Human CGH Array 8 × 60 K; Agilent) with a 41 kb overall median probe spacing (Agilent). Sample and reference DNA (Agilent) samples were labeled with Cy5 and Cy3 dyes (Agilent), respectively. Hybridization was performed according to the manufacturer's protocols. Arrays were scanned using the G2565BA DNA Micro-array Scanner (Agilent Technologies) and data were extracted using Feature Extraction Software v10.7 (Agilent Technologies). The Aberration Detection Method 2 (ADM-2) algorithm in Cytogenomics v5.0 (Agilent Technologies) was used to identify contiguous genomic regions corresponding to chromosomal aberrations or copy number variations, which were scored based on the average quality weighted log ratio of the sample and reference channels.

**Fluorescence in situ hybridization**. The EWSR1 break-apart (Kreatech), *EWSR1-FLI1* dual-fusion (Cytocell) and *BCR-ABL1* dual-fusion FISH probe (Kreatech) FISH probes were used to detect t(11;22) and t(9;22) chromosomal translocations. Tissue sections (5 mm) were deparaffinized in xylene and rehydrated in ethanol (Sigma-Aldrich). Tissue sections were pre-treated in 2-[N-morpholino] ethane-sulphonic acid (DAKO), followed by pepsin digestion (DAKO). After dehydration, the samples were denatured in the presence of the appropriate probe at 66 °C for 10 min and left overnight for hybridization at 37 °C in a hybridizer machine (DAKO). The slides were then washed with 20xSSC/Tween20 buffer at 63 °C and mounted on fluorescence mounting medium (with DAPI). FISH signals were manually scored by counting the number of nuclei with split signals across the tissue. FISH images were captured using a CCD camera (Photometrics SenSys camera) connected to a PC running the Zytovision image analysis system (Applied Imaging Ltd., UK).

**Targeted deep sequencing**. Potential off-target sites were predicted using the CRISPR Design web server (http://crispr.mit.edu)[53], and Cas-OFFinder (http://www.rgenome.net/cas-offinder/)[59]. Customized PCR primers for the on-target and top-50 potential off-target sites to amplify the sgRNA target region were ordered from IDT with rhAmpSeq index sequences (Supplementary Table 1). Each locus was individually amplified using genomic DNA extracted from the relevant cells. First PCR was performed rhAmpSeq Library Kit (IDT) under the following conditions: 95 °C for 10 min, 10 cycles of 95 °C for 15 s and 61 °C for 4 min and 99.5 °C for 15 min. PCR products of all potential off-target loci were subsequently equally pooled and purified with Agencourt AMPure XP beads (Beckman). The PCR products were then subjected to a second PCR with Illumina indexed adaptor primers under the following PCR conditions: 95 °C for 3 min, 18 cycles of 95 °C for 15 s, 60 °C for 30 s and 72 °C for 30 s, and 72 °C for 1 min. Second PCR products were purified with Agencourt AMPure XP beads (Beckman) at a PCR product:bead ratio of 1:1. The purified libraries were quantified and sequenced using a 300-bp paired-end cycle on the Illumina MiSeq sequencing platform. Results were analyzed with CRISPR RGEN Tools Cas-Analyser software[60].

**Animal experiments**. All experiments with mice conformed to Animal Welfare guidelines and were performed in accordance to protocols approved by the Ethics Committee of the Instituto de Salud Carlos III. Tumors were generated in 8-week-old athymic nude (Hds:Athymic Nude-Foxn1 nu) female mice (Charles River) by subcutaneous implantation of $7.5 \times 10^5$ cells in BD Matrigel (volume ratio 1:1). PDX models were established from patient biopsies (HSJD-ES-006, HSJD-ES-013, HSJD-ES-018) at Sant Joan de Déu Hospital (Barcelona, Spain). Cells were implanted into the flanks of 8-week-old athymic nude mice. Tumor growth was measured blinded to the experimental conditions at the indicated time intervals. Tumor volume was estimated using a caliper and calculated as volume (mm$^3$) = (length [mm]) × (width [mm])$^2$ × 0.52. When tumors reached ~150 mm$^3$, mice were separated blinded to the experimental groups and treated accordingly. Adenoviral inoculation consisted of a total of $2.5 \times 10^9$ pfu (100 μL total volume) administered in 3/4 doses every 3 days. Two doses, one per week, of 1.5 mg/kg of doxorubicin ([DOX], S1208, Selleckchem) were administered in xenografted mice. In some experiments, when tumors reached 1500 mm$^3$ in size, mice were euthanized and tumors were surgically excised and processed for histopathology.

**Adenovirus production**. Viral vectors and viruses were constructed and produced by the Universitat Autonoma de Barcelona (UAB) viral vector production unit. Briefly, homologous recombination in *E. coli* between shuttle plasmids and full-length adenovirus backbones (E1-deleted) were used for the generation of recombinant adenoviral genomes. The adenovirus genomes were transfected to generate a recombinant adenoviral vector in permissive human cells, and purified by two sequential CsCl gradient centrifugations and subjected to chromatography to eliminate the CsCl and exchange buffers. Viral titers were calculated and were between $10^{11}$ and $10^{12}$ particles/ml. Finally, the viral stocks were characterized through the quantification of viral particle content and infectivity. Viral aliquots were stored at −80 °C before injection.

**Electroporation**. The Neon Transfection System (Thermo Fisher Scientific) was used for cell electroporation[34]. Confluent K562 cells were passed into fresh media 24 h prior to electroporation at a low density; the next day cells were resuspended in T solution and electroporated using the established conditions by the manufacturer: 10 μL tips were used to electroporate $4 \times 10^5$ cells with four 10-ms pulses at 1350 V. After electroporation, cells were seeded in 24-well plates containing pre-warmed medium.

**WST-1 measurement**. For the chemotherapy study, 0.05 μg/ml of doxorubicin ([DOX], S1208, Selleckchem), and 0.105 μg/ml of imatinib ([IM], S2475, Selleckchem) were used for in vitro assays of Ewing sarcoma and CML, respectively. WST-1 measurements were performed according to the standard protocol of the manufacturer. Briefly, A673 or K562 cells (10,000/well) were cultured in 96-well plates and overnight. The next day, doxorubicin or imatinib were added to A673 and K562 cells, respectively, and plates were incubated for 24 h. Then 10 μl of

WST-1 was added and plates were incubated for a further 3 h at 37 °C in the incubator. The absorbance was monitored at 450 nm.

**Statistical analysis**. Statistical analysis was performed using GraphPad Prism software package (version 7.0, GraphPad Software). Data from three or more independent experiments were analyzed by two-tailed Student's unpaired *t*-test. Statistical details of the experiments can be found in the corresponding figure legends. To flag levels of significance, asterisks were used as follow (*) *p*-value less or equal than 0.05, (**) *p*-value ≤ 0.01, and (***) *p*-value ≤ 0.001 and (ns) was used when non-significant differences were observed.

**Reporting summary**. Further information on research design is available in the Nature Research Reporting Summary linked to this article.

## Data availability

The NGS data have been deposited in the BioProject database under the accession code PRJNA659633. All other data supporting the findings of this study are available within the article and its Supplementary Information files and from the corresponding author upon reasonable request.

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

## Acknowledgements

This work was supported by grants from the Spanish National Research and Development Plan, Instituto de Salud Carlos III, and FEDER (PI17/02303 and DTS19/00111 to S.R-P.); AEI/MICIU EXPLORA Project BIO2017-91272-EXP and CaixaImpulse (CI18-00017;FuGe) to S.R-P. RT-R. is supported by a postdoctoral fellowship from the Asociación Española Contra el Cáncer (AECC). P.M. is supported by the European Research Council (CoG-2014-646903 and PoC-2018-811220), the Spanish Ministry of Science, Innovation and Universities (SAF2016), and the Catalunya Government (SGR330 and PERIS 2017). J.C.S. is supported by the Spanish Ministry of Science, Innovation and Universities (SAF2017-84248-P) and the Spanish Cell Therapy cooperative research network (TERCEL)(RD16/0011/0011). C.B. is supported by the AECC, Beca FERO, and the ISCIII/FEDER (PI17/01028). P.M. also acknowledges the financial support from the Obra Social La Caixa-Fundaciò Josep Carreras. P.M. is an investigator of the Spanish Cell Therapy cooperative research network (TERCEL). A.M.C. acknowledges funding from ISCIII-FEDER (CP13/00189) and Xarxa de Bancs de Tumors de Catalunya (XBTC; sponsored by Pla Director d'Oncologia de Catalunya). We thank Dr Kenneth McCreath for critical reading of the paper.

## Author contributions

R.T-R. and S.R-P. conceived the project, designed and performed experiments, analyzed the data and wrote the manuscript. M.M-L., did the experiments, analyzed results, and wrote the paper. P.M-G., P.P-S., O.Q-B., S.G-S.; P.P.; M.C.M and F.J.M. assisted in the experiments and collected data. C.B.; H.P.; J.C.S. and P.M. analyzed the data and commented on the paper. All authors read and approved the final manuscript. A.M.C. and J.M. provide the PDX tumors and commented the paper.

## Competing interests

A patent has been filed relating to the data presented in this research study by S.R-P., R.T-R. and M.M-L. (EP18382746.8). The remaining authors declare no competing interests.
