## [Peer Review File · Nature Communications]

REVIEWERS' COMMENTS:

Reviewer #1 (Remarks to the Author):

The authors have significantly improved the manuscript in the revision. In particular, the authors added new data showing combined CRISPR-mediated deletion of EWSR1-FLI1 with doxorubicin inhibits tumor growth in xenograft mouse models (new Fig. 5).

All my questions have been adequately addressed.

Reviewer #4, Reviewer replacement for Reviewer #2 (Remarks to the Author):

Martinez-Lage and colleagues describe a CRISPR/Cas9 based strategy to target introns of fusion oncogenes in a therapeutic context. They show that this strategy can reduce tumor burden and prolong survival in transplant models of EWSR1-FLI1 driven Ewing Sarcoma or BCR-ABL initiated CML.

In their revised manuscript, the authors address a number of critical points raised by the reviewers. However, the major (very fundamental) criticism was not addressed convincingly. All three reviewers stressed in one or the other way the rather limited novelty and advance of the study at the technical and conceptual level.

The concept of targeting fusion oncogenes by CRISPR has been published in an – I find overrated - earlier study. The authors cite this study. Although the approach presented here is clearly superior, it is predictable and does not address the critical bottleneck in the struggle to bring CRISPR closer to applications in human cancer therapy. The critical bottleneck is not the efficacy of gene targeting by CRISPR (there is plenty of technology available), but the delivery of CRISPR components, the repeated applicability etc, which would be needed for effective long term cancer control.

Two aspects raised by the 3 reviewers are also critical to me (and have been sufficiently addressed):

First, additional advanced models that accurately reflect the therapeutic challenges of the human disease would be needed, particularly autochthonous models and models displaying cancer metastasis. The authors state that this is beyond the scope of the study, but I disagree with them. It's metastasis that is killing the patient.

Second, therapeutic effects presented are only short lived. Without showing that the approach can be used to achieve durable therapeutic success, the study remains preliminary to me. It does not sufficiently support the notion that this approach has therapeutic potential in humans.

Reviewer #5, Reviewer Replacement for Reviewer #3 (Remarks to the Author):

I did not review the first version of this manuscript, but at the request of the editors of Nature Communications, I have examined both the original and revised versions of this study, the previous reviewers' comments, and the authors' rebuttal.

Overall, the authors have taken the previous reviewers' comments seriously and have performed relevant experiments to address these comments. For example, to address the comments of Reviewer 1, the authors have added data showing the effects of targeting the EWS-FLI1 locus and treatment with doxorubicin. The authors have also included analysis of genomic inversions and have expanded the text explaining their rationale for employing an adenoviral vector system. To address the comments raised by Reviewer 2, the authors have included data examining genomic deletion rates over time, have assessed additional controls, and have revised the text to clarify some points and enhance its readability. Reviewer 3 also requested clarification as to why the authors selected to use an adenoviral vector, and the revised text addresses this satisfactorily. Reviewer 3 raised the critical issue of the induction of mutations that the authors may not detect. The authors have addressed aspects of this issue more fully in the revised text (page 7 – Supplementary Table 3 and supplementary Fig 4h). However, I am less convinced by the utility of the data presented in Supplementary Fig 4b discussed in the text as follows: "To assess possible side-effects that could affect proliferation of healthy cells, we treated hMSCs with LVCas9_NT or LVCas9_EF, finding no difference in proliferation rates in long-term culture, suggesting no production of collateral cancer-driver genomic alterations (Supplementary Fig. 4b)." I am not convinced that a cell-based experiment conducted over 30 days can prove – "no production of collateral cancer-driver genomic alterations." This reviewer recommends, the authors tone-down this assertion and stress somewhere in the text of the manuscript the limitations of this experiment and that future studies will need to address this question in more detail. To address other comments made by Reviewer 3, the authors have included additional histological data and further analysis of the relative ratio of modified and modified cells over time in relevant experiments. The authors have also described the potential for developing their approach for clinical benefit with more circumspection.

As a substitute reviewer, I am reluctant to request any further experimentation. However, if I had reviewed the first version of this manuscript, I would have wanted to see evidence of changes in the edited A673 and RDES cells' expression profile versus controls. The EWS-FLI1 protein drives a very distinctive gene expression signature. Changes in this profile would increase the likelihood that the downstream effects - cell survival and tumor growth - reflect the loss of EWS-FLI1 transcriptional activity and not an off-target effect. If there is an opportunity for the authors to include this data, it would significantly enhance the reader's ability to see that editing of the EWS-FLI1 locus results in direct changes in fusion oncoprotein function.

There are also two points the authors should address in their discussion.

First is the impact of TP53 status on the efficiency of CRISPR-CAS9 editing. Most Ewing sarcomas do not harbor mutations in TP53, but both RD-ES and A673 cells are TP53 mutant (RD-ES- TP53 p.Arg273Cys (c.817C>T); A673 - Homozygous for TP53 p.Ala119fs*5 (c.354_355insCA)). The presence of wild-type TP53 can negatively affect the efficiency of CRISPR-CAS9 gene editing, and the authors should include a discussion of this issue, and the possible impact this could have on taking their strategy further. If I had reviewed the original version of this study, I would have requested an analysis of the effects of the authors' gene editing strategy in a TP53-wild-type Ewing sarcoma cell line. However, under these circumstances, it is my recommendation that the authors only discuss this limitation of their study.

Second, the authors should also comment on the possible impact of the disruption of other DNA damage response and DNA repair pathways observed in Ewing sarcoma, and potentially, other

fusion-driven cancers. For example, for Ewing sarcoma, the authors should discuss recent findings that EWS-ETS fusion proteins can alter DNA repair pathways (e.g., Gorthi et al., *Nature*, 2018, 555, 387).

Additional minor points

Please revise the sentence "In fact, only a few successful anti-cancer drugs have been developed based on the ability to target FOs 38,39, and many of these treatments have mixed outcomes and can result in recurrence and severe side effects 40,41.

This sentence lacks detail and is misleading, particularly in the light of the success of inhibitors that target BCL-ABL or NTRK-fusions. The authors should develop a more specific argument at this point of their discussion that does more than cite two reviews – references 38 and 39, and two papers that do not discuss treatments targeting FOs, but the side-effects of non-targeted chemotherapy agents.

Please cite alternative references for references 9 and 10; e.g.,

Recurrent fusion of TMPRSS2 and ETS transcription factor genes in prostate cancer. Tomlins SA, Rhodes DR, Perner S, Dhanasekaran SM, Mehra R, Sun XW, Varambally S, Cao X, Tchinda J, Kuefer R, Lee C, Montie JE, Shah RB, Pienta KJ, Rubin MA, Chinnaiyan AM. *Science*. 2005 Oct 28;310(5748):644-8. doi: 10.1126/science.1117679. PMID: 16254181

Mod Pathol . 2020 May;33(5):924-932. doi: 10.1038/s41379-019-0417-3. Epub 2019 Dec 2. NTRK gene rearrangements are highly enriched in MLH1/PMS2 deficient, BRAF wild-type colorectal carcinomas-a study of 4569 cases; Chou et al., PMID: 31792356 DOI: 10.1038/s41379-019-0417-3

Please cite the following review along with reference 31 when discussing different EWS-FLI1 fusions; Sankar and Lessnick, Promiscuous partnerships in Ewing's sarcoma 2011 Jul;204(7):351-65. doi: 10.1016/j.cancergen.2011.07.008. PMID: 21872822 PMID: PMC3164520 (current reference 43). The authors may also wish to add a couple of further references reporting the complexity of EWS-FLI1 fusions at this point in the manuscript.

COMMENTS to REVIEWERS

For easier reading, the reviewers' questions are in Times New Roman Latin typestyle and responses are in Arial type. Questions are copy-pasted from the *Nature Communications* Editor's decision correspondence. Changes in the text of the revised manuscript are indicated by labelling in yellow.

Reviewer #1:

- *The authors have significantly improved the manuscript in the revision. In particular, the authors added new data showing combined CRISPR-mediated deletion of EWSR1-FLII with doxorubicin inhibits tumor growth in xenograft mouse models (new Fig. 5).*

All my questions have been adequately addressed.

We appreciate the kind comment from Reviewer#1.

Reviewer #4, Reviewer replacement for Reviewer #2:

Martinez-Lage and colleagues describe a CRISPR/Cas9 based strategy to target introns of fusion oncogenes in a therapeutic context. They show that this strategy can reduce tumor burden and prolong survival in transplant models of EWSR1-FLII driven Ewing Sarcoma or BCR-ABL initiated CML.

- *In their revised manuscript, the authors address a number of critical points raised by the reviewers. However, the major (very fundamental) criticism was not addressed convincingly. All three reviewers stressed in one or the other way the rather limited novelty and advance of the study at the technical and conceptual level. The concept of targeting fusion oncogenes by CRISPR has been published in an - I find overrated - earlier study. The authors cite this study. Although the approach presented here is clearly superior, it is predictable and does not address the critical bottleneck in the struggle to bring CRISPR closer to applications in human cancer therapy. The critical bottleneck is not the efficacy of gene targeting by CRISPR (there is plenty of technology available), but the delivery of CRISPR components, the repeated applicability etc, which would be needed for effective long term cancer control.*

Two aspects raised by the 3 reviewers are also critical to me (and have been sufficiently addressed):

- *First, additional advanced models that accurately reflect the therapeutic challenges of the human disease would be needed, particularly autochthonous models and models displaying cancer metastasis. The authors state that this is beyond the scope of the study, but I disagree with them. It's metastasis that is killing the patient.*
- *Second, therapeutic effects presented are only short lived. Without showing that the approach can be used to achieve durable therapeutic success, the study remains preliminary to me. It does not sufficiently support the notion that this approach has therapeutic potential in humans.*

The reviewer raises key points shared by the authors and referees #1 and #3. It is well recognized that delivery remains a performance bottleneck in CRISPR-based therapy development. In our previous revision, we expanded the discussion with different delivery systems for CRISPR, such as viral and non-viral platforms that could be exploited in the future to deliver the FO-targeting system to tumor cells. Finally, following the editor's suggestion, we have tone down the claims on the therapeutic application of our gene targeting approach.

Reviewer #5, Reviewer Replacement for Reviewer #3:

I did not review the first version of this manuscript, but at the request of the editors of Nature Communications, I have examined both the original and revised versions of this study, the previous reviewers' comments, and the authors' rebuttal.

Overall, the authors have taken the previous reviewers' comments seriously and have performed relevant experiments to address these comments. For example, to address the comments of Reviewer 1, the authors have added data showing the effects of targeting the EWS-FLI1 locus and treatment with doxorubicin. The authors have also included analysis of genomic inversions and have expanded the text explaining their rationale for employing an adenoviral vector system. To address the comments raised by Reviewer 2, the authors have included data examining genomic deletion rates over time, have assessed additional controls, and have revised the text to clarify some points and enhance its readability. Reviewer 3 also requested clarification as to why the authors selected to use an adenoviral vector, and the revised text addresses this satisfactorily. Reviewer 3 raised the critical issue of the induction of mutations that the authors may not detect. The authors' have addressed aspects of this issue more fully in the revised text (page 7 - Supplementary Table 3 and supplementary Fig 4h).

- *However, I am less convinced by the utility of the data presented in Supplementary Fig 4b discussed in the text as follows: "To assess possible side-effects that could affect proliferation of healthy cells, we treated hMSCs with LVCas9_NT or LVCas9_EF, finding no difference in proliferation rates in long-term culture, suggesting no production of collateral cancer-driver genomic alterations (Supplementary Fig. 4b)." I am not convinced that a cell-based experiment conducted over 30 days can prove - "no production of collateral cancer-driver genomic alterations." This reviewer recommends, the authors tone-down this assertion and stress somewhere in the text of the manuscript the limitations of this experiment and that future studies will need to address this question in more detail.*

We thank the reviewer for mentioning this point. We have eliminated the sentence "suggesting no production of collateral cancer-driver genomic alterations" from the manuscript (Page 7, line 14), and we have included a comment regarding the limitation of the experiment in the Discussion section (Page 14, line 31).

To address other comments made by Reviewer 3, the authors have included additional histological data and further analysis of the relative ratio of modified and modified cells over time in relevant experiments. The authors have also described the potential for developing their approach for clinical benefit with more circumspection.

As a substitute reviewer, I am reluctant to request any further experimentation. However, if I had reviewed the first version of this manuscript, I would have wanted to see evidence of changes in the edited A673 and RDES cells' expression profile versus controls. The EWS-FLI1 protein drives a very distinctive gene expression signature. Changes in this profile would increase the likelihood

that the downstream effects - cell survival and tumor growth - reflect the loss of EWS-FLII transcriptional activity and not an off-target effect. If there is an opportunity for the authors to include this data, it would significantly enhance the reader's ability to see that editing of the EWS-FLII locus results in direct changes in fusion oncoprotein function.

We thank the reviewer for this excellent suggestion. Unfortunately, we are not able to include this assay in the present study.

There are also two points the authors should address in their discussion.

- *First is the impact of TP53 status on the efficiency of CRISPR-Cas9 editing. Most Ewing sarcomas do not harbor mutations in TP53, but both RD-ES and A673 cells are TP53 mutant (RD-ES- TP53 p.Arg273Cys (c.817C>T); A673 - Homozygous for TP53 p.Ala119fs*5 (c.354_355insCA)). The presence of wild-type TP53 can negatively affect the efficiency of CRISPR-CAS9 gene editing, and the authors should include a discussion of this issue, and the possible impact this could have on taking their strategy further. If I had reviewed the original version of this study, I would have requested an analysis of the effects of the authors' gene editing strategy in a TP53-wild-type Ewing sarcoma cell line. However, under these circumstances, it is my recommendation that the authors only discuss this limitation of their study.*

We agree with the referee about the potential impact of TP53 status on the efficiency of our strategy. Recent studies have raised concerns regarding the performance of CRISPR-Cas9 screens using TP53 wild-type human cells due to a p53-mediated DNA damage response (DDR) limiting the efficiency of generating viable edited cells (doi.org/10.1038/s41591-018-0049-z; doi.org/10.1038/s41591-018-0050-6).

Although we did not include the description of the TP53 status in our original manuscript, the PDX models were characterized by NGS once established. Two of them are TP53 wild-type, and the third one is a mutant. The results from our xenograft and PDX studies suggest that the TP53 status does not affect the efficiency of our approach. These data have not been included in the manuscript because the studies, conducted by Dr. Carcaboso, are still on-going.

- *Second, the authors should also comment on the possible impact of the disruption of other DNA damage response and DNA repair pathways observed in Ewing sarcoma, and potentially, other fusion-driven cancers. For example, for Ewing sarcoma, the authors should discuss recent findings that EWS-ETS fusion proteins can alter DNA repair pathways (e.g., Gorthi et al., Nature, 2018, 555, 387).*

We have included a comment regarding those issues in the Discussion section (Page 13, line 33).

- *Additional minor points Please revise the sentence "In fact, only a few successful anti-cancer drugs have been developed based on the ability to target FOs 38,39, and many of these treatments have mixed outcomes and can result in recurrence and severe side effects 40,41.*

This sentence lacks detail and is misleading, particularly in the light of the success of inhibitors that target BCL-ABL or NTRK-fusions. The authors should develop a more specific argument at this point of their discussion that does more than cite two reviews ??? references 38 and 39, and two papers that do not discuss treatments targeting FOs, but the side-effects of non-targeted chemotherapy agents.

Following the reviewer's suggestion, we have revised the sentence (Page 13, line 14).

- *Please cite alternative references for references 9 and 10; e.g., Recurrent fusion of TMPRSS2 and ETS transcription factor genes in prostate cancer. Tomlins SA, Rhodes DR, Perner S, Dhanasekaran SM, Mehra R, Sun XW, Varambally S, Cao X, Tchinda J, Kuefer R, Lee C, Montie JE, Shah RB, Pienta KJ, Rubin MA, Chinnaiyan AM. Science. 2005 Oct 28;310(5748):644-8. doi: 10.1126/science.1117679. PMID: 16254181 Mod Pathol . 2020 May;33(5):924-932. doi: 10.1038/s41379-019-0417-3. Epub 2019 Dec 2. NTRK gene rearrangements are highly enriched in MLH1/PMS2 deficient, BRAF wild-type colorectal carcinomas-a study of 4569 cases; Chou et al., PMID: 31792356 DOI: 10.1038/s41379-019-0417-3*

Following the reviewer's suggestion, we have included the alternative references in the manuscript (Page 3, line 28).

- *Please cite the following review along with reference 31 when discussing different EWS-FLII fusions; Sankar and Lessnick, Promiscuous partnerships in Ewing's sarcoma 2011 Jul;204(7):351-65. doi: 10.1016/j.cancer.2011.07.008. PMID: 21872822 PMCID: PMC3164520 (current reference 43). The authors may also wish to add a couple of further references reporting the complexity of EWS-FLII fusions at this point in the manuscript.*

Following the reviewer's suggestion, we have included the new references in the manuscript (Page 5, line 16).